# Usable Information and Evolution of Optimal Representations During Training

**Michael Kleinman**[1]  **Alessandro Achille**[2]  **Daksh Idnani**[1]  **Jonathan C. Kao**[1]
[1]University of California, Los Angeles   [2]Caltech
`{michael.kleinman,dakshidnani}@ucla.edu` `aachille@caltech.edu`
`kao@seas.ucla.edu`

## Abstract

We introduce a notion of usable information contained in the representation learned by a deep network, and use it to study how optimal representations for the task emerge during training. We show that the implicit regularization coming from training with Stochastic Gradient Descent with a high learning-rate and small batch size plays an important role in learning minimal sufficient representations for the task. In the process of arriving at a minimal sufficient representation, we find that the content of the representation changes dynamically during training. In particular, we find that semantically meaningful but ultimately irrelevant information is encoded in the early transient dynamics of training, before being later discarded. In addition, we evaluate how perturbing the initial part of training impacts the learning dynamics and the resulting representations. We show these effects on both perceptual decision-making tasks inspired by neuroscience literature, as well as on standard image classification tasks.

## 1   Introduction

An important open question for the theory of deep learning is why highly over-parametrized neural networks learn solutions that generalize well even though the model can in principle memorize the entire training set. Some have speculated that neural networks learn minimal but sufficient representations of the input through implicit regularization of Stochastic Gradient Descent (SGD) (Shwartz-Ziv & Tishby, 2017; Achille & Soatto, 2018), and that the minimality of the representations relates to generalizability. Follow-up work has disputed the validity of some of these claims when using deterministic deep networks (Saxe et al., 2018), leading to an ongoing debate on the notion of optimality of representations and how they are learned during training.

Part of the disagreement stems from the use of information-theoretic quantities: most previous studies in deep learning have analyzed the amount of information that the learned representation contains about the inputs using Shannon's mutual information. However, when the mapping from input to representation is deterministic, the mutual information between the representation and input is degenerate (Saxe et al., 2018; Goldfeld et al., 2018). Rather than study the mutual information in a neural network, here we instead define and study the "usable information" in the network, which measures the amount of information that can be extracted from the representation by a learned decoder, and is scalable to high dimensional realistic tasks. We use this notion to quantify how relevant and irrelevant information is represented across layers of the network throughout the training process, and how this is affected by the optimization algorithms and the network pretraining.

In particular, we propose to study a simple task inspired by decision-making tasks in neuroscience, where inputs and outputs are carefully designed to probe specific information processing phenomena. We then extend our findings to standard image classification tasks trained with state-of-the-art models. Our neuroscience-inspired task is the checkerboard (CB) task (Chandrasekaran et al., 2017; Kleinman et al., 2019). In the CB task, one discerns the dominant color of a checkerboard filled with red and green squares. The subject then makes a reach to a left or right target whose color matches the dominant color in the checkerboard (Fig 1a). This task therefore involves making two binary choices: a color decision (i.e., reach to the red or green target) and a direction decision (i.e., reach to left or right). Critically, the color of the targets (red left, green right; or green left, red right) is

random on every trial. The direction decision output is conditionally independent of the color decision, as detailed further in Fig 1b and Section B.6, even though the color information needs to be used to solve the task. This task allows us to evaluate how both of these components of information are represented through training and across layers.

We used this task and extensions to study the evolution of minimal representations during training. If a representation is sufficient and minimal, we refer to this representation as optimal (Achille & Soatto, 2018). Our contributions are the following. **(1)** We introduce a notion of usable information for studying representations and training dynamics in deep networks (Section 3). **(2)** We used this notion to characterize the transient training dynamics in deep networks by studying the amount of usable relevant and irrelevant information in deep network layers and across training epochs. We first use the CB task to gain intuition of the training dynamics in a simplified setting. We find that training with SGD is critical to bias the network toward learning minimal representations in intermediate layers (Section 4.1). This adds to the literature suggesting that SGD results in minimal representations of input information (Achille & Soatto, 2018; Shwartz-Ziv & Tishby, 2017) while avoiding some of the pitfalls. **(3)** We used the intuition gained from the simple task, evaluating our findings on CIFAR-10 and CIFAR-100 task using modern architectures. Remarkably, we find that the networks increased usable information about an irrelevant component of information early in training and discarded it later on in training to arrive at a minimal sufficient solution, consistent with a proposed (Shwartz-Ziv & Tishby, 2017) though controversial theory (Saxe et al., 2018).

## 2 RELATED WORK

Some efforts to understand why neural networks generalize focus on representation learning, that is, how deep networks learn optimal (i.e., minimal and sufficient) representations of inputs in order to solve a task. Typically, representation learning is focused on studying the properties of the asymptotic representations after training (Achille & Soatto, 2018). Recent work suggests that these asymptotic representations contain minimal but sufficient input information for performing a task (Achille & Soatto, 2018; Shwartz-Ziv & Tishby, 2017). Implicit regularization coming from SGD, and in particular from the use of large learning rates and small batch sizes, is believed to play an important role in forming these minimal sufficient representations.

How does the training process lead to these minimal but sufficient asymptotic representations? Shwartz-Ziv & Tishby (2017) propose that there are two distinct phases of training: an empirical risk minimization phase where the network minimizes the loss on the training set, and a "compression" phase where the network discards information about the inputs that do not need to be represented to solve the task. Recently, Saxe et al. (2018) challenged this view, arguing that the observed compression was dependent on the activation function and the mutual information estimator used in Shwartz-Ziv & Tishby (2017). These works highlight the challenges of estimating mutual information to study how representations emerge through training.

In general, estimating mutual information from samples is challenging for high-dimensional random variables (Paninski, 2003). The primary difficulty in estimating mutual information is estimating a high-dimensional probability distribution from the samples, since generally the number of samples required scales exponentially with the dimension. This is impractical for realistic deep learning tasks where the representations are high dimensional. To estimate the mutual information, Shwartz-Ziv & Tishby (2017) used a binning approach, discretizing the activations into a finite number of bins. While this approximation is exact in the limit of infinitesimally small bins, in practice, the size of the bin affects the estimator (Saxe et al., 2018; Goldfeld et al., 2018). In contrast to binning, other approaches to estimate mutual information include entropic-based estimators (e.g., Goldfeld et al. (2018)) and a nearest neighbours approach (Kraskov et al., 2004). Although mutual information is difficult to estimate, it is an appealing quantity to summarily characterize key aspects of the transient neural network training behavior because of its invariance to smooth and invertible transformations. In this work, rather than estimate the mutual information directly, we instead define and study the "usable information" in the network, which corresponds to a variational approximation of the mutual information (Barber & Agakov, 2003; Poole et al., 2019) (see Sections 3 and A.1). Recently, such variational approximations to mutual information have been viewed as a meaningful characterization of representations in deep networks, and the theoretical underpinnings of this approach are beginning to be investigated (Xu et al., 2020; Dubois et al., 2020).

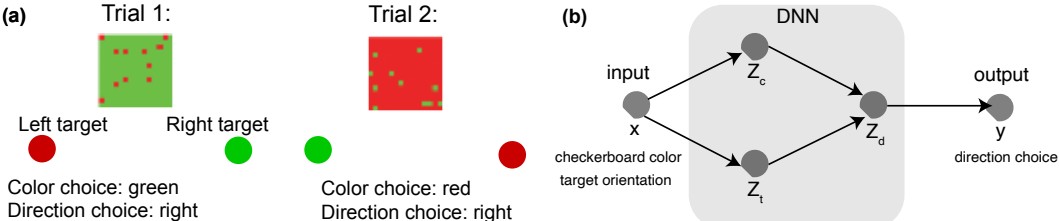

Figure 1: **(a)** Checkerboard task. Given two binary target locations (left or right) with randomly selected binary colors (red or green), one has to discern the dominant color in the checkerboard and reach to the target of the dominant color. On every trial, there is a correct color and direction choice. However, the identities of the left and right targets are random every trial, decoupling the direction and color decision. **(b)** We trained a deep neural network to perform the task by specifying the proportion of green and red squares on the checkerboard, as well as two scalars denoting the colors of the left and right target. The network was trained to output the correct direction choice. As only the direction, but not the color choice, was reported, given a representation of the correct direction choice $Z_d$, the network does not need to represent the color choice $Z_c$ in deeper layers. $Z_t$ is the representation of the target orientation.

Research into the training dynamics of deep networks, and how they represent relevant and irrelevant task information, is nascent. A related study by Achille et al. (2019) found that early periods of training were critical for determining the asymptotic network behavior. Additionally, it was found that the timing of regularization was important for determining asymptotic performance (Golatkar et al., 2019), with regularization during this "critical period" having the most influential effect. Notably, both of these studies found an initial increase in the amount of information that weights encode about the dataset (as measured by the Fisher information), that coincides with the critical period of learning. This phase is followed later in training by a "forgetting" phase where the network discards unnecessary information. This suggests that a similar dynamic to the one we study can be observed in weight space instead of representation space.

## 3 USABLE INFORMATION IN A REPRESENTATION

A deep neural network consists of a set of $\ell$ layers, with each layer forming a successive representation of the input. A representation $Z_\ell$ may store information in a variety of ways. It may be that a complex transformation is required to read out the information, or it may be that a simple linear decoder could read out the information. In both cases, from an information-theoretic perspective, the same information is contained in the representation, however, there is an important distinction regarding how "usable" this information is. Information is usable if later layers, which comprise affine transformations and element-wise nonlinearities, can easily extract it to solve the task. Equivalently, usable information should be decodable by a separate neural network also employing affine transformations and element-wise nonlinearities.

Formally, we define the usable information that a representation $Z$ contains about a quantity $Y$, which may refer to the output or a component of the input, as:

$$I_u(Z; Y) = H(Y) - L_{CE}(p(y|z), q(y|z)). \tag{1}$$

Here, $H(Y)$ is the entropy, or uncertainty, of $Y$, and $L_{CE}$ is the cross-entropy loss on the test set of a discriminator network $q(y|z)$ trained to approximate the true distribution $p(y|z)$. Our definition is motivated in the following manner. The test set cross-entropy loss approximates how much uncertainty there is in the output $Y$ given $Z$ and the discriminator. A low loss implies that there is low uncertainty in $Y$ given $Z$, or that the discriminator can extract a lot of "information" about $Y$ from $Z$. If the logarithm in the cross-entropy loss is in base 2, it is measured in bits. If the value of $Y$ were approximately the same for any $Z$, there would be little uncertainty in $Y$ to begin with, so it is important to know the amount of uncertainty in $Y$ given $Z$ with respect to the initial uncertainty in $Y$. What is most relevant is the amount of remaining uncertainty in $Y$ given $Z$. Thus we use the

difference in uncertainty $H(Y) - L_{CE}$ as the amount of "usable information" that $Z$ contains about $Y$, as shown in our definition in Equation 1.

This definition is appealing to study representations, in part, because it can be computed from samples of $Z$ and $Y$, and is a quantity that is comparable across network training. We estimate $L_{CE}$ using a small neural network that learns a distribution $q(y|z)$. To train the network, we sample activations $Z$ and the quantity $Y$ and learn $q(y|z)$ by minimizing the cross-entropy loss on a training set. We then evaluate the $L_{CE}$ on the test set (Equation 1). We provide details about the neural network and the training we used for decoding in Appendix B.3 and C.2. We also show in the Appendix that the usable information is a lower bound on the mutual information (Appendix A.1). Importantly, usable information also is not constrained by the data processing inequality; that is, the information can be made more "usable" by transformation to later layers, consistent with the representation learning view that later layers are forming improved representations of the inputs (Xu et al., 2020).

## 4 EXPERIMENTS

Our goal was to characterize how optimal representations are formed through SGD training. We trained multiple network architectures on tasks and assessed the usable information in representations across layers and training epochs. For a given architecture and task, all hyper-parameters were kept constant throughout experiments, unless explicitly stated.

To develop intuition, we initially investigate how small fully connected networks represent the relevant and irrelevant information in the CB Task. We trained two different network architectures, 'Small FC': 5 layers, with $10 - 7 - 5 - 4 - 3$ units in each layer, 'Medium FC': $100 - 20 - 20 - 20$. Small FC was a network used in prior literature (Shwartz-Ziv & Tishby, 2017; Saxe et al., 2018). Our networks were fully-connected and used ReLU activation. We trained the networks using SGD with a constant learning rate to perform the CB task, described in detail in Appendix B.4. The hyper-parameters used for the CB experiments are listed in Appendix B.5.

In our CB task experiments, we quantified the usable color and direction information in the hidden representation, $Z_\ell$. In the $n = 2$ CB task, the color information represents half of the input information. We emphasize that, unless otherwise specified, the network was only trained to output the correct direction choice, so given a representation of the direction, representing the color choice is irrelevant. Therefore, a minimal representation should not include information about the color choice, since it is not necessary to represent given a representation of the direction decision. To make the task more complex, we also generalized the CB task to have $n = 10$ and $n = 20$ targets.

We then use this framework to examine how relevant and irrelevant information are represented in more realistic tasks and architectures, and how hyper-parameters affect the learning dynamics. We define a coarse labelling of task labels and study how the network represents the fine and coarse labelling through training, using a ResNet-18 (He et al., 2016) and All-CNN (Springenberg et al., 2015) on CIFAR-10 and CIFAR-100.

### 4.1 SGD WITH RANDOM INITIALIZATION RESULTS IN MINIMAL SUFFICIENT REPRESENTATIONS IN THE CB TASK

We first assessed the optimality of the network representations by training Small FC networks on the CB task using $n = 2$ colors (Fig 2a) using a random initialization for the weights. In particular, the initial weights do not contain information about the dataset. We computed the usable color and direction information across layers of the neural network and epochs of training. In our plots, later layers are denoted by darker shades. In deeper layers, there was a decrease in usable color information, corresponding to more minimal representations. After training, the asymptotic representation in the last layer contained zero usable color information and 1 bit of usable direction information. To visualize this minimal sufficient representation, we plotted the activations of the 3 units in the last layer of the Small FC network for different inputs. These visualizations are labeled by the correct color (red and green) and direction (cross or circle). In the asymptotic representation, representation of the input color is overlapping (red and green), while the representation of the direction output is separable (crosses and circles), forming a minimal sufficient representation.

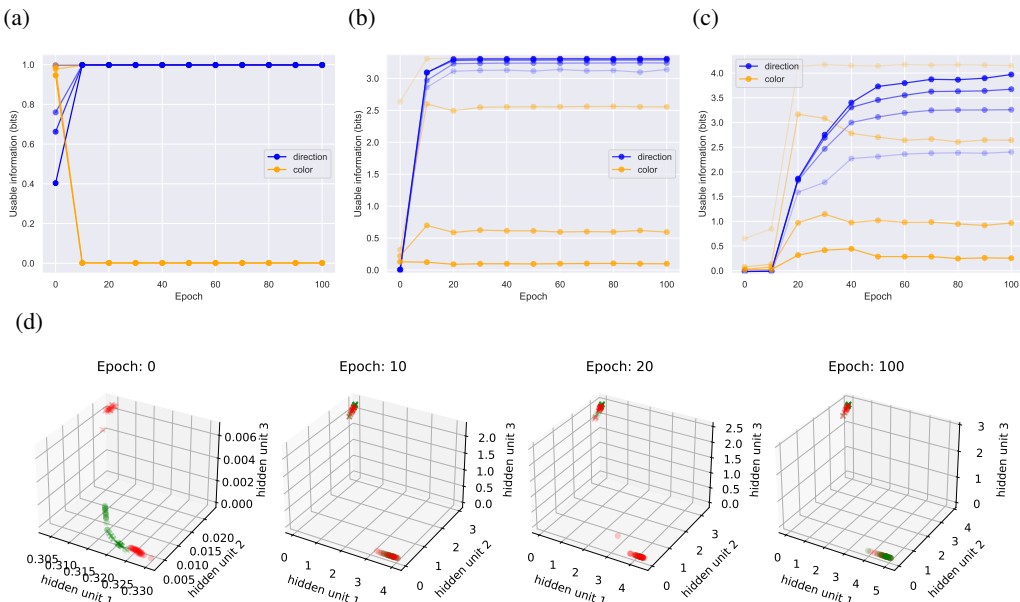

Figure 2: **SGD with random initialization leads to minimal representations. (a)** Small FC network trained on the $n = 2$ checkerboard task. Max usable direction and color information: 1 bit. This network was trained without regularization for 100 epochs using SGD with a learning rate of 0.05 and batch size of 32. Blue (orange) lines correspond to usable information about the direction (color) decision in the representation. Darker shades of color correspond to deeper layers in the network. In the asymptotic representations, we observed that direction information was high across layers, while color information decreased in the later layers. The usable color information was approximately zero in the last layer of the Small FC network. **(b)** Medium FC network trained with $n = 10$ checkerboard colors. Max usable direction and color information: 3.32 bits. In the last layer, there is nearly zero usable color information. Across layers, there is a decrease in usable color information, and an increase in usable direction information. **(c)** Medium FC network trained with $n = 20$ checkerboard colors, a batch size of 128 and a learning rate of 0.5. Max usable direction and color information: 4.32 bits. In the later layers (darker shades) there is small usable color information, but large usable direction information. **(d)** Visualization of the activations of the last layer of Small FC from (a) at epochs [0, 10, 20, 100], where the correct color choice is denoted by the marker color (red or green) and the correct direction choice is denoted by marker shape (crosses or dots). After training the crosses and dots are overlapping, corresponding to nearly zero usable color information and nearly 1 bit of direction information. This is a minimal and sufficient representation to solve the task.

To test if this observed minimality was a result of our simple task, we extended the CB task to a variant with $n$ input checkerboard colors, with $n$ corresponding output direction classes. We trained networks using a larger architecture (Medium FC). We show results for $n = 10$ and $n = 20$ classes in Fig 2b,c. We observed similar phenomena to the $n = 2$ case: there was decreasing usable color information in deeper layers, and nearly zero color information in the last layer's representation. In contrast, there was significant usable direction information across all layers in the asymptotic representation, with usable information about the direction increasing for deeper layers. We validated our results using different random initializations (Figures 9, 10, 11).

These results show that, for a simple task with SGD and random initialization, minimal sufficient representations emerge through training. Asymptotic representations were sufficient to perform the task, but contained less usable color information in deeper layers, approaching zero color information in the last layer. In this simple task, we observed that it was possible for the network to solve the task with nearly zero usable color information in its last layer across training (Fig 2b,c).

We also examined how changing the initialization by pretraining the network to output the color choice affected the resulting representations. We found that the resulting representations were not

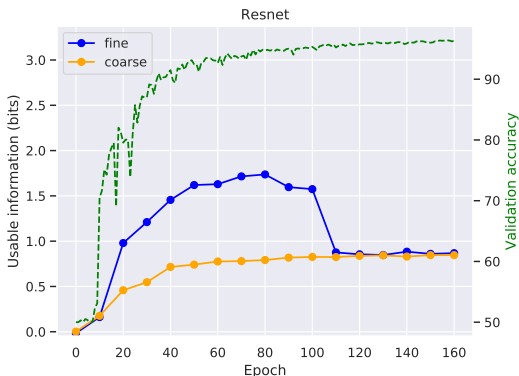

Figure 3: **Usable fine and coarse class information in a ResNet-18 on CIFAR-10.** The fine classes (show in blue) correspond to the 10 CIFAR-10 classes. The coarse classes correspond a superclass consisting of all the even and odd classes. We trained the network to output the correct coarse class, which corresponds to 1 bit of information. Through training epochs, while the validation accuracy (green dashed line) is increasing, the information about the coarse class also increases towards 1 bit. Early in training, the usable information about the fine label also increased, even though the network was not explicitly provided any information about the fine class. Around epoch 100, the network "forgets" this fine label information. The scale of the validation accuracy is shown on the right hand side of the plot.

minimal for the $n = 2$ checkerboard case (Fig 6a), retaining some structure from the initialization (Fig 6d). This result also held for the CB task with $n = 10$ and $n = 20$ (Fig 6b,c). Furthermore, we found that pretraining on the color choice led to worse generalization performance (Fig 7).

## 4.2 ACQUISITION AND FORGETTING OF USABLE INFORMATION IN MODERN DEEP NETWORKS

Using a similar approach as we did for the CB task to characterize relevant and irrelevant information, we next investigated how modern deep neural networks trained with SGD learned task representations. To study learning dynamics, we investigated (1) how networks learned and represented task information as well as information about a representative semantically meaningful variable, and (2) how this information was represented across training epochs. To this end, we defined coarse labels corresponding to groups of classes in the CIFAR-10 and CIFAR-100 datasets. The CIFAR-100 dataset defines fine labels corresponding to each of the 100 classes, as well as 20 coarse labels corresponding to meaningful groupings of 5 from the 100 classes. In the CIFAR-10 case, we defined two coarse labels arbitrarily, corresponding to even and odd class labels. Thus, when training the network to output the coarse label, we can investigate the network's representation of the semantically meaningful fine label description, which serves as a proxy for the computation and representations that the network is learning. We note that, when trained to output the coarse label, a minimal representation should contain no additional information about the fine label.

We trained a ResNet-18 (He et al., 2016) to output the coarse label of CIFAR-10, using an initial learning rate of $0.1$ with exponential annealing $(0.97)$, momentum $(0.9)$, and a batch size of $128$. We investigated the usable information in the last layer of the ResNet-18, which has a dimension of $512$. We found that while training the network to predict the coarse-grained class, the network acquired information about the coarse-grained class, evidenced by an increase in usable information during training (orange curve) while validation accuracy (green dashed line; scale on the right hand side of plot) was increasing (Fig 3). Strikingly, while the validation accuracy and usable coarse-grained class information increased, the information about the fine labels first increased and then decreased (around epoch 100). It then decreased to minimality, storing no additional usable information about the fine labels than was contained in the coarse labels. These learning dynamics were proposed Shwartz-Ziv & Tishby (2017), but due to controversies of their information estimation and experimental setup, have been widely debated (Saxe et al., 2018). We emphasize that even though we did

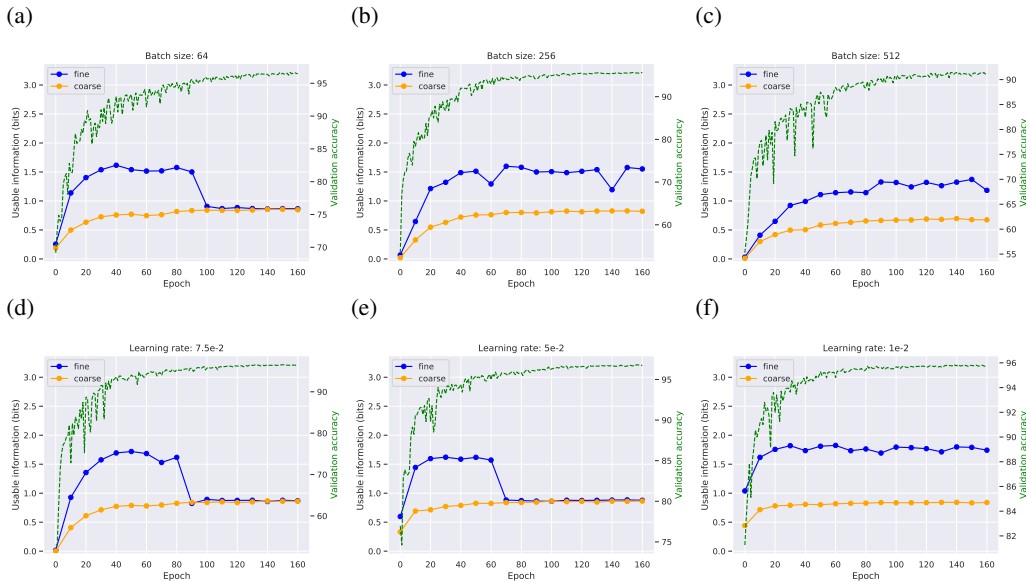

Figure 4: **Sensitivity to hyper-parameters.** **(a-c)** The usable coarse and fine label information through training with a batch size of 64, 256, and 512 (a batch size of 128 was used in Fig 3. The learning dynamics only undergo a compression at small batch sizes of 128 or less. The validation accuracy is higher for smaller batch sizes as well. The plot of a batch size of 1024 is in Fig 8. **(d-f)** Usable coarse and fine label information using initial learning rates of 0.075, 0.05 and 0.01 (a learning rate of 0.1 was used in Fig 3. With larger learning rates, the network observed an increase and decrease in fine label information. With a smaller learning rate 0.01, the network exhibited an increase in fine label information, without a subsequent decrease. The final validation accuracies (green dashed lines) are approximately comparable (96.5%, 96.8% and 95.8% respectively) though lowest with initial learning rate of 0.01 when the network did not form a minimal representation.

not ask the network to acquire information about the fine labels, SGD naturally led the network to learn information about the fine label, and then decreased this information later in training.

Together, these results show that SGD tends to result in minimal representations, which may be guided by interesting learning dynamics. To achieve this minimality, the network displays a learning motif where it learns additional information early in training, then discards it later on. We next investigate how these findings depend on hyper-parameter choices, architecture, and task.

### 4.3 SENSITIVITY OF USABLE INFORMATION TRAINING DYNAMICS TO HYPER-PARAMETERS, ARCHITECTURE, AND TASK

Using this framework, we evaluated how hyper-parameter choices affected the learning dynamics in deep networks. We focus on the ResNet-18 trained on CIFAR-10 in Figure 3. We varied the batch sizes from 64 to 1024 and found that a small batch size led to dynamics similar to that of Fig 3, while a larger batch size did not lead to minimal representations (Fig 4a-c). Results for a batch size of 1024 are shown in Fig 8. The learning rate also affected the learning dynamics. We found that all networks increased the information about the fine labels during training. However, we found that only for large initial learning rates did the network "forget" the superfluous information. Results for a learning rate of 0.001 are also shown in the appendix in Fig 8. We found that small learning rates (0.001) or large batch sizes (512 or larger) led to lower validation accuracy. Thus, the implicit regularization coming from the use of SGD with a small batch size and large learning rate, which is common in practical settings, is crucial for learning minimal sufficient representations. Here we have provided an underpinning for these choices by exposing their associated learning dynamics.

Additionally, we investigated whether the phenomenon of acquiring "superfluous" task information was common across different architectures and tasks. We used an All-CNN (Springenberg et al., 2015) trained on CIFAR-10 to output the binary coarse label, observing a similar trend with an

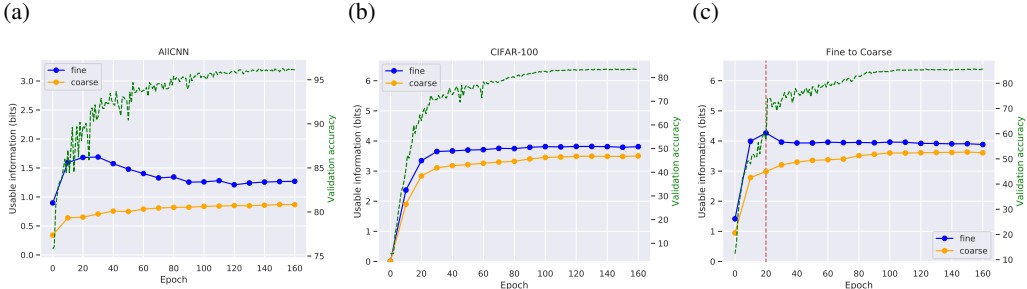

Figure 5: **Different Architecture, Task, and learning schedule (a)** Using an All-CNN architecture Springenberg et al. (2015), we observe a similar trend in the learning dynamics of usable information, with a increase and decrease in the fine label information during the CIFAR-10 task. This decrease does not lead to a completely minimal representation, though it does become close to minimal. **(b)** We trained a ResNet-18 on the coarse labels in the CIFAR-100 task, and tracked the information the network had about the fine and coarse label through training. We find that the network converges to an approximately minimal representation, though it did not undergo a noticeable increase and decrease in the fine label information, suggesting that this learning motif depends on the structure of the task. **(c)** Pretraining the network to output the fine labels before epoch 20 led to improved final performance ($85.6\%$ vs $83.5\%$) in **(b)**. Note that the validation accuracy for the first 20 epochs was the validation accuracy on the 'fine' labels task, and was the validation accuracy on the 'coarse' task after epoch 20.

increase and decrease in the usable information about the fine label (Fig 5a). In this case, the information about the fine label did not decrease to minimality, but nonetheless, there was a significant reduction in the fine label information, suggesting that SGD naturally compresses additional input information. Finally, we evaluated how a ResNet-18 represented task information using the CIFAR-100 dataset. This dataset is accompanied with 100 fine labels and 20 coarse labels, corresponding to groupings of the 100 classes. We used the same hyper-parameters as in Fig 3. We trained the network to output the coarse labels, observing an increase to approximately 3.5 bits of usable information. The network achieved a nearly minimal representation (Fig 5b).

It is important to note that for this setting of hyper-parameters in the CIFAR-100 task (the same as in the CIFAR-10 case), SGD did not show a visible increase followed by a decrease in usable information in the fine labels, a result different than what we observed in CIFAR-10. We conjectured this could be due to at least three potential reasons: (1) the hyper-parameter settings may be suboptimal, which we observed may result in learning dynamics that do not increase then decrease fine information (Fig 4c, f). (2) In CIFAR-100, coarse and fine labels are semantically similar, so there may not be not much more information to be naturally learned in the fine than the coarse labels, and further that it is possible that while the information about the fine labels remains approximately flat, the network is forgetting information about aspects of the fine labels while learning other parts of fine label information in the process of increasing coarse label information and arriving at a nearly minimal representation. (3) CIFAR-100 has relatively few examples, 500 per fine label, impacting the learning of fine label information. Despite these limitations, our results from CIFAR-10 suggest that SGD learning dynamics that increase then decrease information about the fine label should result in more optimal representations and higher validation accuracy. To test this, we performed an experiment where we pretrained the network to output fine label information until epoch 20, after which the network then was trained to output coarse information. This training process resulted in learning dynamics that resembled SGD learning in Fig 3. We observed that these learning dynamics resulted in networks with a $2.1\%$ increase in validation accuracy (compare Fig 5b and c). These results support that learning dynamics that increase, and then decrease, information about inputs, may result in more optimal representations that achieve higher validation accuracy.

## 5 DISCUSSION

We introduced a notion of the usable information in the representation, which reflects the amount of information that can be extracted by a learned decoder, for understanding the training dynamics

in deep networks. This definition is appealing, in part, due to its flexibility. For instance, if it is important to understand how accessible the information is to a linear decoder, it suffices to apply our formulation of usable information using a linear decoder trained with cross-entropy loss. In contrast, if the goal is to extract all information present in a representation, regardless of how accessible this information is, one can train a high capacity nonlinear decoder. Since neural networks are powerful function approximators, as the function approximation improves, the decoder will approach the optimal decoder. In this case, the usable information approaches Shannon mutual information, as the lower bound becomes tight (Section A.1). Future theoretical and empirical work should investigate the tightness of this bound and its dependence on training parameters.

In our case, we used a relatively small nonlinear neural network as the decoder, which provided insight into the evolution of optimal representations through training on simple tasks inspired by neuroscience literature and on image classification tasks. These tasks allowed us to show that the implicit regularization of SGD plays an important role in learning minimal sufficient representations. In particular, in standard hyper-parameter settings, we observed learning dynamics where the network learns to encode semantically meaningful but ultimately irrelevant information early in training, before later discarding this information to arrive at a minimal sufficient representation.

Monkeys performing the checkerboard task, like our networks, also had minimal sufficient representations in an output (motor) area (Chandrasekaran et al., 2017; Kleinman et al., 2019). Despite the obvious implementation differences of both information processing systems, we speculate that the general effects coming from a noisy learning process, which led to minimal sufficient representations in our artificial networks, may be an important factor leading to minimal sufficient representations in biological networks.

It is remarkable that in the CIFAR-10 task, SGD naturally exploited the semantically meaningful structure of the fine labels, in order to solve the coarse labels task. In general, it is difficult to identify the features that are being learned during training, and whether they correspond to something semantically meaningful. However by defining a coarse label, our task setup allowed us to study how semantically meaningful information was represented during training. During training, the network increased the information about the semantically important part of the input, even when only asked to output the coarse label. It then decreased the information later in training. We did not notice such a major increase in CIFAR-100, perhaps due to the nature of the dataset or hyper-parameter configuration. However, by inducing the network to follow similar learning dynamics to Fig 3 by pretraining the network to output the fine labels, we were able to improve the performance on the coarse labelling task. This suggests that a detailed understanding of the training dynamics and the features learned is important for learning optimal representations and successfully transferring representations between tasks.

Using usable information, we observed an increase and decrease in the information about an irrelevant variable, which has been proposed (Shwartz-Ziv & Tishby, 2017), but has been debated, largely due to controversies over the estimation of Shannon's mutual information (Saxe et al., 2018). Our observation is in accordance with the ideas of Shwartz-Ziv & Tishby (2017), and importantly we have observed these dynamics on modern architectures and realistic tasks. Our results are also consistent with a complementary view of information in the weights, where it has been observed that the Fisher Information increased and decreased during training (Achille et al., 2019), corresponding to a critical period in neural network training.

## ACKNOWLEDGMENTS

We thank the anonymous reviewers for their thoughtful feedback and suggestion of new experiments. MK was supported by the National Sciences and Engineering Research Council (NSERC). JCK was supported by an NSF CAREER Award 1943467, a UCLA Computational Medicine Amazon Web Services Award, and a NVIDIA GPU Grant.

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

# A  PROOFS

## A.1  USABLE INFORMATION LOWER BOUNDS THE MUTUAL INFORMATION

The entropy of a distribution is defined as

$$H(x) = \mathbb{E}_{x \sim p(x)} \left[ \log \frac{1}{p(x)} \right]. \tag{2}$$

The mutual information, $I(X;Y)$, can be written in terms of an entropy term and a conditional entropy term:

$$I(Z;Y) = H(Y) - H(Y|Z). \tag{3}$$

We want to show that:

$$I(Z;Y) \geq I_u(Z;Y) := H(Y) - L_{CE}(p(y|z), q(y|z)) \tag{4}$$

It suffices to show that:

$$H(Y|Z) \leq L_{CE} \tag{5}$$

where $L_{CE}$ is the cross-entropy loss on the test set. For our study, $H(Y)$ represented the known distribution of output classes, which in our case were equiprobable.

$$H(Y|Z) := \mathbb{E}_{(z,y) \sim p(z,y)} \left[ \log \frac{1}{p(y|z)} \right] \tag{6}$$

$$= \underbrace{\mathbb{E}_{(z,y) \sim p(z,y)} \left[ \log \frac{1}{q(y|z)} \right]}_{\text{cross-entropy loss}} - \underbrace{\mathbb{E}_{z \sim p(z)} \left[ \mathrm{KL}(p(y|z)||q(y|z)) \right]}_{\geq 0}, \tag{7}$$

$$\leq \mathbb{E}_{(z,y) \sim p(z,y)} \left[ \log \frac{1}{q(y|z)} \right] := L_{CE} \tag{8}$$

To approximate $H(Y|Z)$, we first trained a neural network with cross-entropy loss to predict the output, $Y$, given the hidden activations, $Z$, learning a distribution $q(y|z)$. The KL denotes the Kullback-Liebler divergence. We multiplied (and divided) by an arbitrary variational distribution $q(y|z)$ in the logarithm of equation 6, leading to equation 7. The first term in equation 7 is the cross-entropy loss commonly used for training neural networks. The second term is a KL divergence and is therefore non-negative. In our approximator, the distribution $q(y|x)$ is parametrized by a neural network. When the distribution $q(y|z) = p(y|z)$, our variational approximation of $H(Y|Z)$, and hence approximation of $I(Z;Y)$ is exact (Barber & Agakov, 2003; Poole et al., 2019).

# B  ADDITIONAL RESULTS AND DETAILS IN THE CHECKERBOARD TASK

## B.1  SGD WITH NON-RANDOM INITIALIZATION MAY NOT FORM MINIMAL REPRESENTATIONS IN THE CB TASK

Implicit regularization in SGD is hypothesized to result in a minimal representation through compression of irrelevant input information, also called a "forgetting" phase (Shwartz-Ziv & Tishby, 2017; Achille & Soatto, 2018; Achille et al., 2019). We tested this hypothesis by initializing networks with significant color information, and subsequently performing SGD on the CB task. We then evaluated whether SGD resulted in networks with minimal color representations. We initialized the weights by pretraining the network to output the color decision for 20 epochs, which required the network to represent color information. After 20 epochs, we reverted to training on the CB task, where only the direction decision was reported. Since the learning rate was kept constant, the pretrained weights can be viewed as a different initialization in parameter space for the modified task.

Strikingly, we found that the resulting representations were not minimal for the $n = 2$ checkerboard case (Fig 6a). This result also held for the CB task with $n = 10$ and $n = 20$ (Fig 7b,c). While we observed some compression of usable color information through training, the asymptotic representations had significantly greater than zero color information. In Fig 7b, we observed all layers

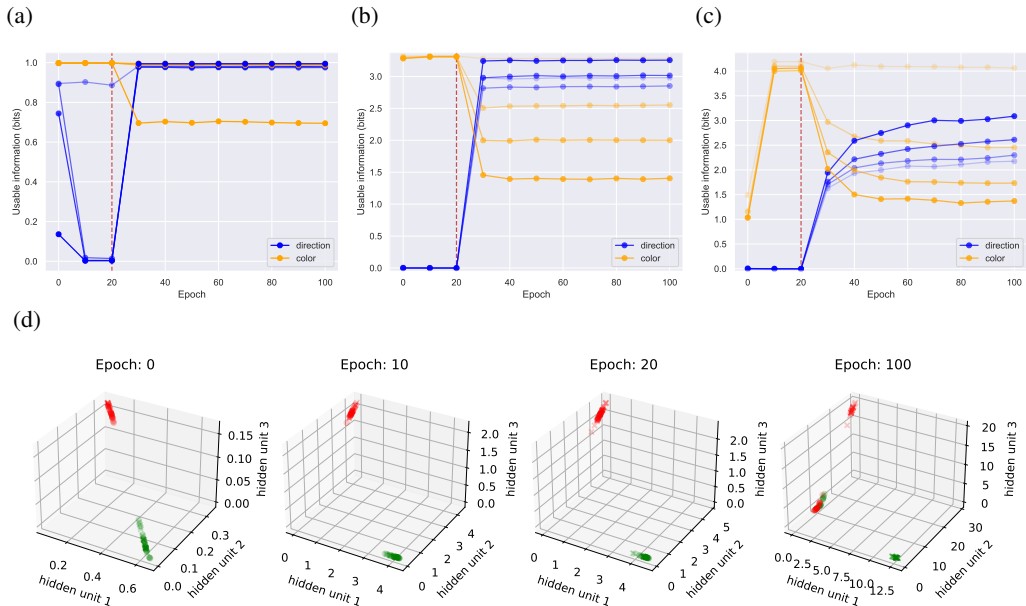

Figure 6: Usable color and direction information in a network through training following pretraining the network to output color, not direction. Pretraining occurred for the first 20 epochs, indicated by the dashed red line. Subsequently, the network was trained to output direction, as in Fig 2. **(a)** Usable information for Small FC trained on the $N = 2$ CB task. Usable color information increased in training, and decreased when the loss function changed. However, the asymptotic representation is not minimal. **(b)** Medium FC trained on $N = 10$ CB task. Similarly, the network formed a representation of color during pretraining, but the asymptotic representation is not minimal. **(c)** Medium FC trained on $N = 20$ checkerboard task. **(d)** Visualization of the Small FC network in (a) showing that an optimal representation is not formed. The asymptotic representation in the last area has separate representations for red and green crosses. These should be overlapping in a minimal representation.

had more usable color information than the direction information in the first layer. The network therefore solved the task using an alternative representation that was not minimal. We visualized the activations corresponding to the asymptotic non-minimal representations of Small FC in Fig 6d. In the early epochs the red and green points converge (both crosses and dots) as a result of successful pretraining. However, when we trained the CB task starting at epoch 20, the representations changed. While the dot clusters for red and green checkerboards are overlapping, the cross clusters are not. This representation is not minimal as color information can be decoded above chance.

These results show that the initialization affects the asymptotic representation of neural networks. SGD, under particular initializations, may not lead to minimal representations of the task inputs. This suggests there is a trade-off between learning a minimal representation and simply reusing the existing representations present in the initial weights. Initial structure in the network representations from pretraining, such as the separation of the red and green crosses in the last layer representation, was maintained even when performing SGD to train a different task. Together, these results suggest that while SGD compresses representations towards minimality, it finds a solution that is functionally related to the initial representation. This may correspond to a optima in the neighborhood of the initialization.

## B.2 RELATIONSHIP BETWEEN PRETRAINING, MINIMALITY, AND GENERALIZATION IN THE CB TASK

Our results show that the minimality of network representations, and therefore solutions, depends on initialization. All trained networks (for $n$ larger than 2), however, achieved zero training er-

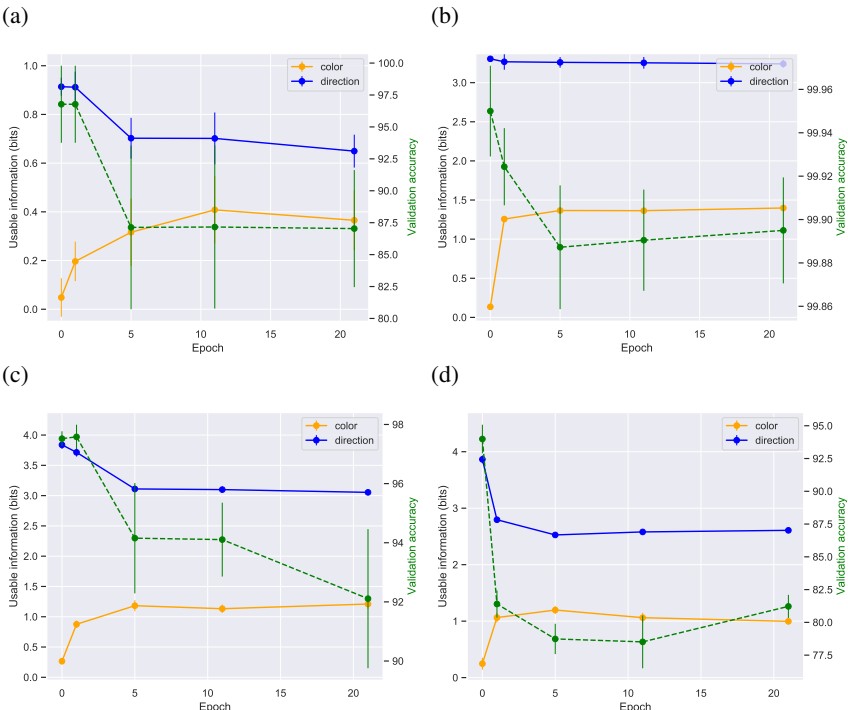

Figure 7: **(a)** Final usable information and validation accuracy (green dashed line) as a function of pretraining epoch for the CB task ($n = 2$) averaged over 8 random initializations. **(b)** Final usable information and accuracy as a function of pretraining epoch for the CB task ($n = 10$) averaged over 8 random initializations. **(c)** Final usable information and accuracy as a function of pretraining epoch for the CB task ($n = 20$) averaged over 8 random initializations. **(d)** Final usable information and accuracy as a function of pretraining epoch for the CB task ($n = 25$) averaged over 8 random initializations. Error bars show the S.E.M.

ror. A natural question to ask is how does the pretraining affect the resulting representation and generalization performance?

To answer this, we varied the number of epochs that we pretrained the CB tasks of $n = 2$, $n = 10$, and $n = 20$ classes, and quantified the usable color and direction information, as well as the trained network's test accuracy to understand how the network generalizes (Fig. 7). We found that networks trained with longer pretraining had less minimal representations and worse generalization performance. This was true regardless of the number of classes, but the effect was more pronounced (in terms of absolute difference in accuracy) when the network did not solve the task perfectly without pretraining. We note that regardless of how long the networks were pretrained for, the networks were subsequently trained for the same number of epochs (80), with the same learning rate throughout training. One interpretation is that when using existing structure to solve the task, the network learned a suboptimal solution to solving the task, increasing the chance of overfitting. Another interpretation is that the pretraining changed the distribution of the weights, affecting the minimality and generalization.

## B.3 Details of neural network for usable information in the CB Task

To estimate usable information, we computed the cross-entropy loss of a decoder $q(y|z)$ that predicts $Y$ from $Z$. The decoder was a three-layer neural network, with 128, 64, and 32 units per layer, with Leaky-ReLU activations (slope = 0.2), batch-norm and dropout ($p = 0.7$). At each epoch, 1250 training samples were generated and supplied to the decoder, along with either the corresponding correct direction or color choice. We evaluated the cross-entropy loss on 3750 test samples to minimize overfitting. We trained the network for 100 epochs using a learning rate of 0.5 for 'Medium FC' and 0.05 for 'Small FC.'

### B.4 CHECKERBOARD TASK DESCRIPTION

Following the conventions of Kleinman et al. (2019), we modeled the CB task (Fig 1a), inputting the checkerboard color and target configuration to a neural network that outputted the direction choice (Fig 1b). We minimized the cross-entropy loss of the network output and the ground truth output. We extended the checkerboard task to the $n$ checkerboard task by increasing the number of checkerboards. Each target was 1 out of the $n$ colors, with the targets forming an 'n-polygon'. The correct direction corresponds to the direction of the target having the same color of the checkerboard. We specified the color of each target using a one-hot encoding, and the color of the checkerboard as a one-hot encoding. Noise with mean 0 and standard deviation of 0.1 was added to the checkerboard inputs. The target and checkerboard color inputs were concatenated to form an input vector. The correct direction of the target was the output.

### B.5 DETAILS OF CB EXPERIMENTS

The following are the hyper-parameters used in our experiments. We trained two different network architectures, 'Small FC': 5 layers, with $10 - 7 - 5 - 4 - 3$ units in each layer, 'Medium FC': $100 - 20 - 20 - 20$. We trained networks using SGD with a constant learning rate throughout training.

**FC Small,** $n = 2$:

- batch size: 32, learning rate: 0.05, number of data samples: 10000 (90% train, 10% validation)

**Medium FC,** $n = 10$:

- batch size: 64, learning rate: 0.5, number of data samples: 25000 (90% train, 10% validation)

**Medium FC,** $n = 20$:

- batch size: 128, learning rate: 0.5, number of data samples: 50000 (90% train, 10% validation)

**Medium FC,** $n = 25$:

- batch size: 128, learning rate: 0.5, number of data samples: 75000 (90% train, 10% validation)

### B.6 DEFINITION OF RELEVANT AND IRRELEVANT INFORMATION IN THE CB TASK

In the CB task, the color of the checkerboard and target configuration (inputs) are necessary to determine the correct direction to reach (output). While both a color and direction decision are made, after the direction is determined, the color decision no longer needs to be represented: the network can generate the correct output with only the direction representation. Formally, the output $y$ is conditionally independent of the color representation, $Z_c$, given the direction representation $Z_d$ (i.e., $y \perp\!\!\!\perp (Z_c, Z_t)|Z_d$, as illustrated by the graph in Fig 1b). Hence, given a representation of the direction choice, the color choice (and target configuration) no longer needs to be represented. We emphasize that, in general, the output is not independent of the color representation and target configuration representation $Z_t$, i.e., $y \not\perp\!\!\!\perp (Z_c, Z_t)$, hence information about the dominant color of the checkerboard is necessary to compute $y$. When this conditional independence holds, we call the conditionally independent variable "irrelevant." We therefore refer to the color choice as "irrelevant" and the direction choice as "relevant." We study how these components evolve together throughout training.

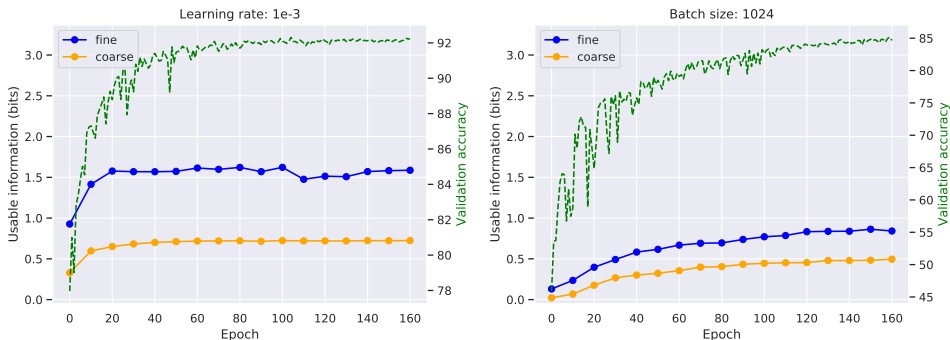

Figure 8: **(Left)** Usable information for initial learning rate of 0.001 in CIFAR-10. The information about the fine labels does not decrease, and the validation accuracy only reaches 92%, in co ntrast to Fig 3 where the validation accuracy reached 96%. **(Right)** Usable information for batch size of 1024 in CIFAR-10.

## C ADDITIONAL RESULTS AND DETAILS IN THE CIFAR-10 AND CIFAR-100 TASK

### C.1 CIFAR-10 AND CIFAR-100 TASK DESCRIPTION

We trained a ResNet-18 and an All-CNN architecture to output a superclass corresponding to the twenty coarse-grained classes in CIFAR-100 and, in CIFAR-10, to an arbitrary superclass corresponding to the even and odd classes. Accordingly, a minimal representation should only encode the superclass.

### C.2 DETAILS OF NEURAL NETWORK FOR USABLE INFORMATION

To estimate usable information, we computed the cross-entropy loss of a decoder $q(y|z)$ that predicts $Y$ from $Z$. We used a two-layer neural network, with 200 and 100 with Leaky-ReLU activations (slope = 0.2), batch-norm and dropout ($p = 0.7$). At each epoch, 7500 samples were supplied to the decoder, along with either the corresponding correct direction or color choice. We evaluated the cross-entropy loss on 2500 test samples. We trained the network for 50 epochs using Adam with a learning rate of 0.01 and weight decay of 0.001.

### C.3 DETAILS OF NEURAL NETWORK TRAINING

In our experiments, unless otherwise stated, we trained a ResNet-18 (He et al., 2016) with an initial learning rate of 0.1 decaying smoothly with a factor of 0.97 at each epoch, batch size of 128, momentum of 0.9 and weight decay with coefficient 0.0005. For the All-CNN (Springenberg et al., 2015) we used a batch size of 128, initial learning rate of 0.05 decaying smoothly by a factor of 0.97 at each epoch, momentum of 0.9, and weight decay with coefficient 0.001. We used standard data augmentation with random translations up to 4 pixels and random horizontal flipping. These parameter configurations were taken directly from prior work (Achille et al., 2019).

## D ADDITIONAL PLOTS

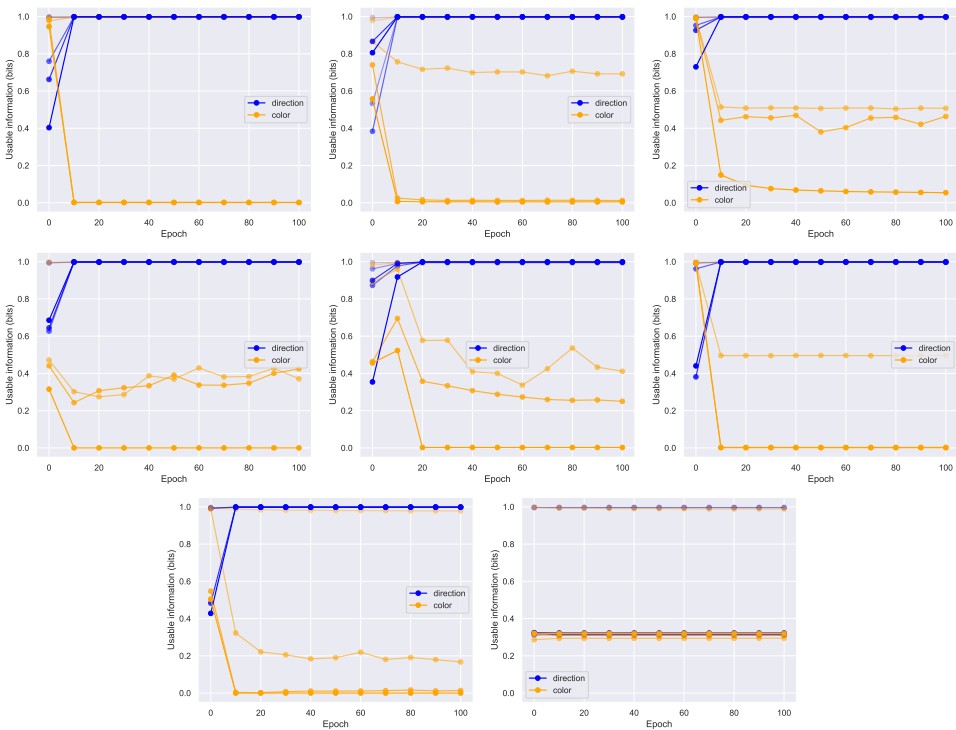

Figure 9: Evolution of usable information for eight random initializations for the $n = 2$ CB task.

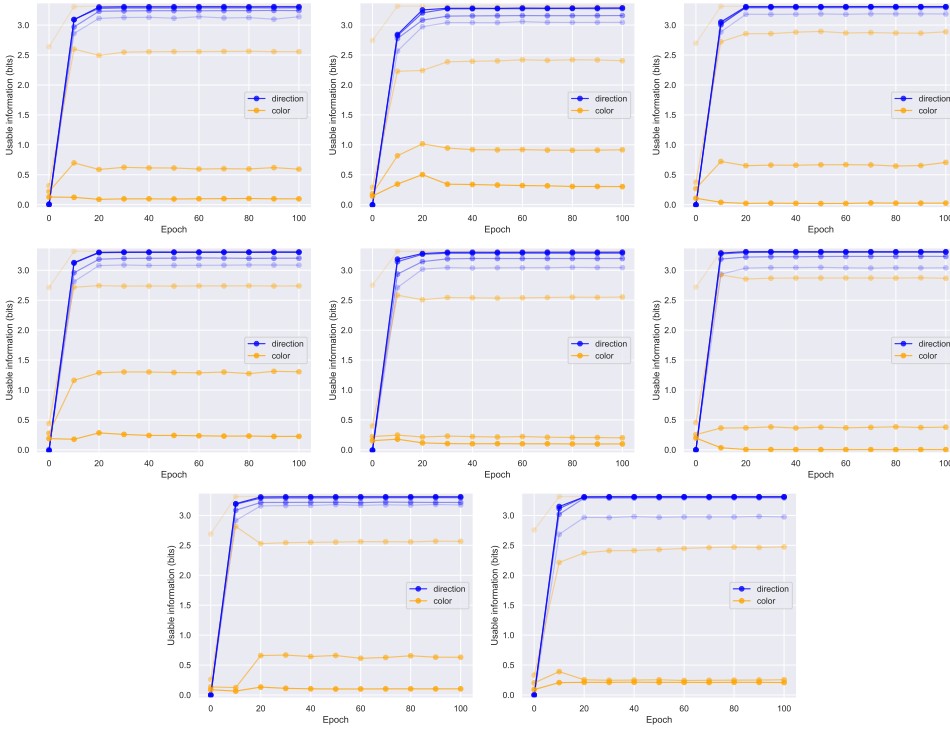

Figure 10: Evolution of usable information for eight random initializations for the $n = 10$ CB task.

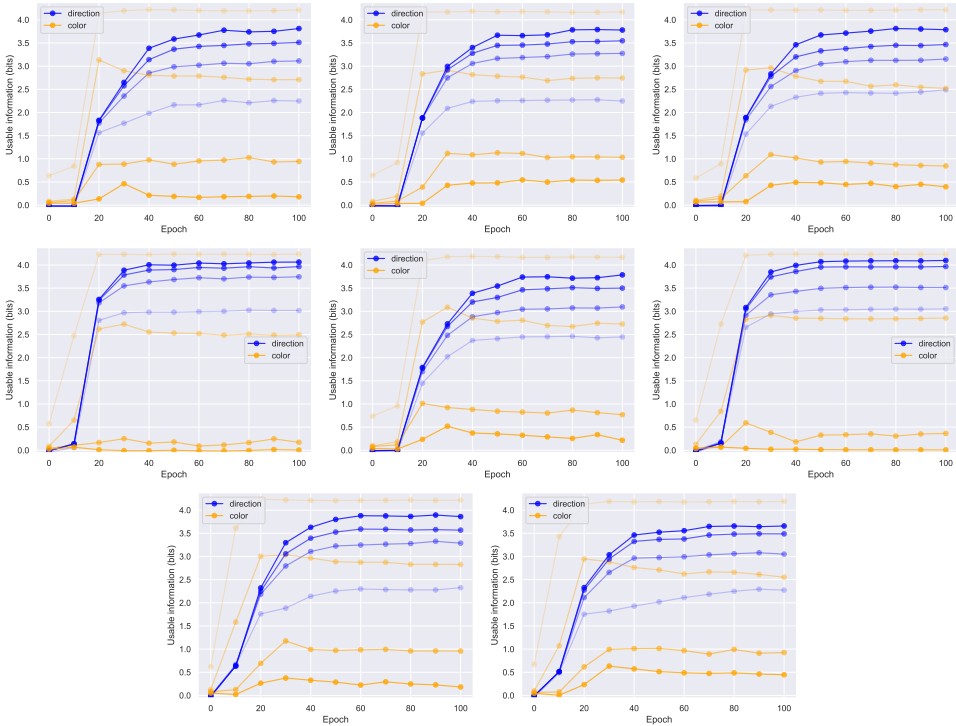

Figure 11: Evolution of usable information for eight random initializations for the $n = 20$ CB task.

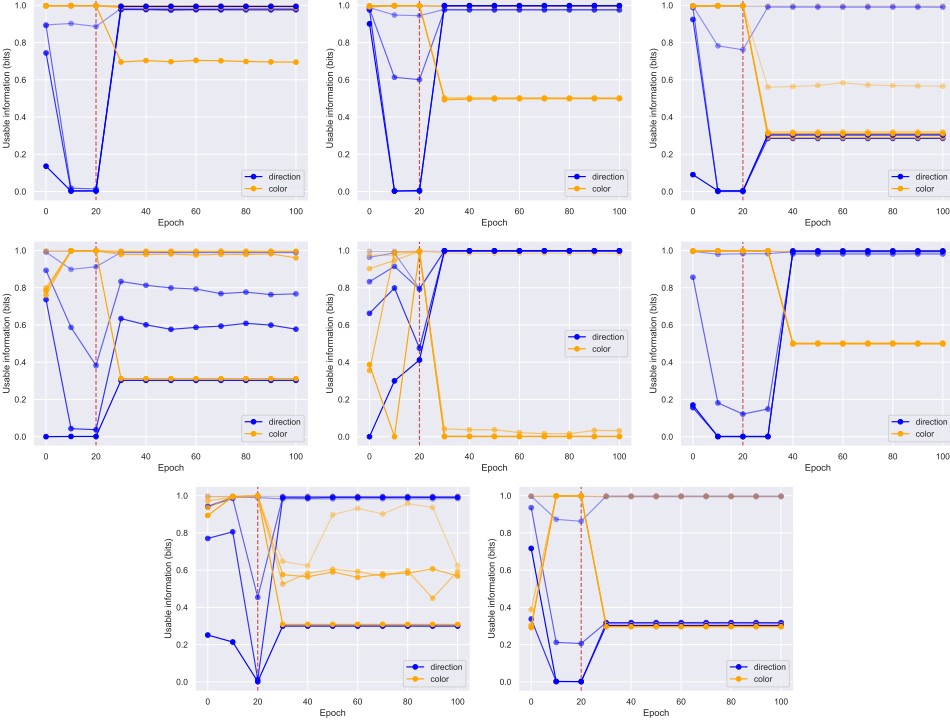

Figure 12: Evolution of usable information for eight random initializations for the $n = 2$ CB task with 20 epochs of pretraining. If the the usable information was negative, indicating that the decoder overfit, we set the usable information to 0. Note that this occurred for a very small number of points.

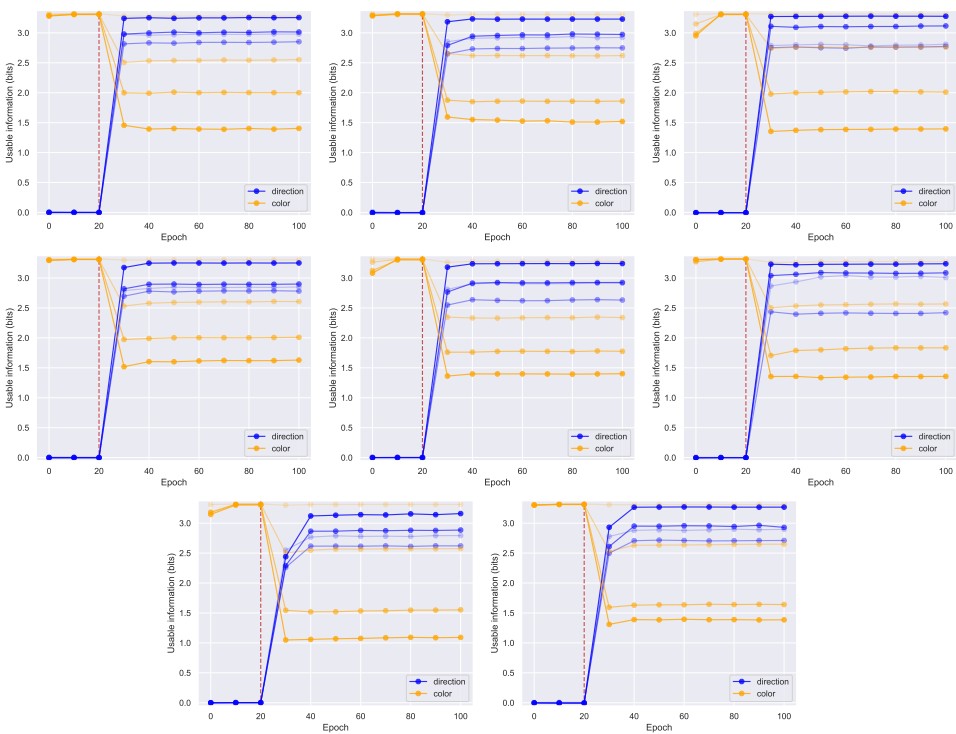

Figure 13: Evolution of usable information for eight random initializations for the $n = 10$ CB task with 20 epochs of pretraining.

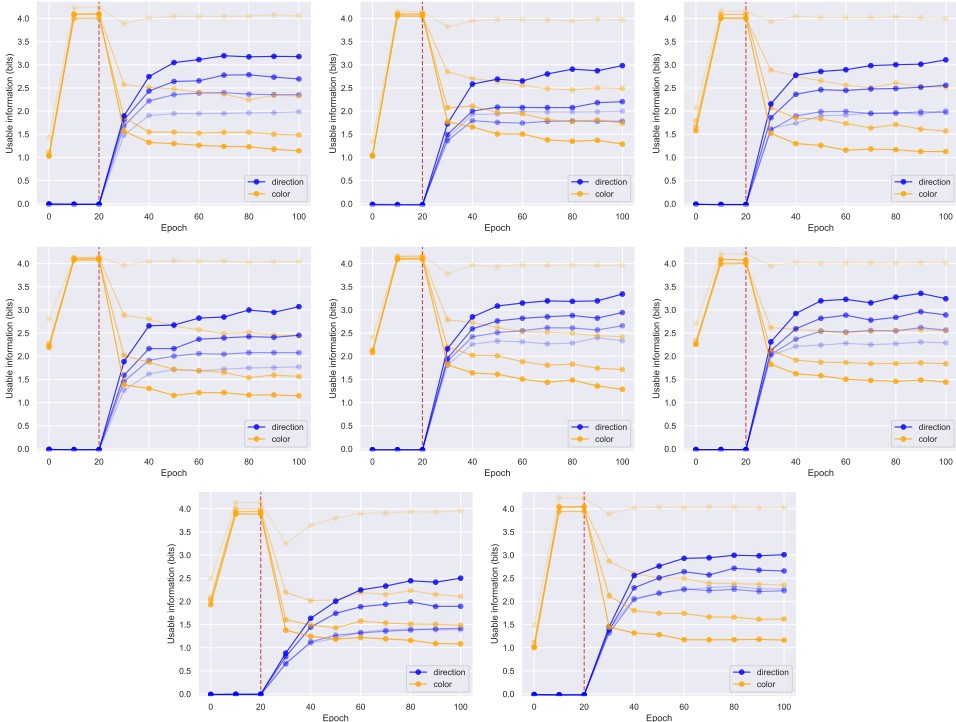

Figure 14: Evolution of usable information for eight random initializations for the $n = 20$ CB task with 20 epochs of pretraining.

