# OpenReview forum: "Usable Information and Evolution of Optimal Representations During Training"
_ICLR.cc/2021/Conference — ICLR 2021 Poster_

### Official Review · AnonReviewer3 · 2020-10-27
**Interesting findings, but not enough evidence of generality**

**Rating:** 7
**Confidence:** 4

**Review:**

The paper studies how initialization and the implicit regularization of SGD affect the training dynamics of neural networks in terms of minimality and sufficiency of learned representations. The main findings are that 1) SGD with random initialization learns almost minimal and sufficient representations and 2) SGD with an initialization that contains information about irrelevant factors fails to converge to minimal representations, increasing the chance of overfitting. These findings are interesting, useful for understanding neural networks, relevant to the ICLR community, but lack evidence of generality.

**Task choices.**
Most of the experiments in this paper are done on the checkerboard task. In this task one is given a checkerboard, each cell of which is either red or green. One of these colors appears more (is dominant) and the task is to decide which color is dominant. The key aspect is that the input also contains 2 targets: left and right, with one being green and one being red. Instead of directly predicting the dominant color, the subject should pick the target whose color matches the dominant color. Importantly, the color arrangement of targets is picked at random. The authors also consider the task of predicting whether an MNIST digit is odd or even. These two tasks are too simple, which restricts the generality of conclusions. I suggest to consider harder tasks, for example classifying CIFAR-100 images, where the target is the superclass (1-20) and the irrelevant factor is the exact class (1-5). Additionally, it would be interesting to consider cases when the training data is such that there is a small mutual information between the irrelevant factor and the target. Will SGD with random initialization find a solution that has even smaller mutual information with irrelevant factors (i.e. sacrificing sufficiency for minimality)?

**Network.**
Throughout the paper only fully connected networks are considered. Additionally, the last hidden layer always has <= 20 units. For generality, it would be better to consider also larger and more modern networks, such as ResNets.

**Activation function.**
Saxe et al. [1] showed that the choice of activation function is crucial when judging about compression in late stages of training. The presented paper can be improved by considering other choices of activation functions.

**The role of SGD.**
The implicit regularization of SGD arises from its stochasticity. The findings of this paper suggest that this stochasticity has a key role in finding minimal representations. This should be verified by comparing to standard gradient descent (i.e. batch size = number of examples).

**Minor comments**
1. The description of the checkerboard task starting at the last paragraph of the first page can be improved.
2. Did you consider using a more powerful decoder? In Fig. 2c for example, we see that later layers have more usable information about the direction. This means that there is a room for strengthening the decoder.

P.S. I am willing to increase the score if the authors address the above concerns about generality.

# Update
Thanks for the rebuttal, it addressed my main concerns. The new results on CIFAR-10 and CIFAR-100 with fine and course labels match the results on the checkerboard task. This increases the generality of the main claims. The new experiments also confirm that the level of noise in SGD has a key role in finding minimal representations. Furthermore, they show that when training with SGD with enough amount of noise, the usable information with fine labels increases initially and then decreases. This improves our understanding of the phenomenon introduced by [2], which was later debated by [1]. For these mentioned reasons, I updated the rating from "5: Marginally below acceptance threshold" to "7: Good paper, accept".

**References**
[1] Saxe, Andrew M., et al. "On the information bottleneck theory of deep learning." Journal of Statistical Mechanics: Theory and Experiment 2019.
[2] Ravid Shwartz-Ziv and Naftali Tishby. Opening the black box of deep neural networks via information. CoRR,  abs/1703.00810, 2017.

---

> ### Author Response · Authors · 2020-11-21
> **Response**
>
> We thank the reviewer for their thoughtful comments and experimental suggestions. We believe that they have led to a significantly improved and more thorough manuscript.
>
> Regarding task choices and network architecture, we now apply our framework to study harder tasks that are of more interest to the community, such as CIFAR-100, and CIFAR-10, using more realistic architectures such as the ResNet and an AllCNN. We discuss this experiment in the updated Sect. 4.2. We observe that convolutional networks trained using standard settings undergo dynamics similar to the one we observe on the simpler checkerboard task. In particular, depending on the choice of hyper-parameters, we observe that the network learns to encode irrelevant information at the beginning of the training, which is later forgotten to achieve minimality of the representation. We have also identified the effect of hyperparameters including batch size and learning rate. We varied the batch size from 64 to 1024, and varied the initial learning rate from 0.1 to 0.001. We find that for large learning rates and small batch sizes that the network converges to a minimal representation. This suggests that the implicit regularization from SGD is a critical factor for these learning dynamics. Additionally, we find that these networks undergo interesting learning dynamics, in that they initially learn information about the irrelevant information, and discard it later in training.
>
> We thank the reviewer for the helpful suggestions regarding these experiments, which better corroborate our findings and make the framework relevant to a larger part of the community. Interestingly, we uncovered interesting dynamics of learning, proposed with a different formalism by Shwart-Ziv and Tisbhy, and challenged by Saxe et al. We found that over a range of standard hyperparameter settings, that the networks increased the irrelevant information early in training, followed by discarding it later on. But crucially, this happens when measuring *usable* information, and may not be the case, as noted by Saxe et al., when measuring Shannon’s mutual information.
>
> We opted to focus our hyperparameter variations on the harder task using ReLU activations. Notably, Saxe et al. claimed that the compression was only due to the saturation from using tanh nonlinearities, thus we wanted our analyses to focus on the more relevant use of ReLU nonlinearity for which Saxe did not observe compression.
>
> Minor Comments:
>
> 1. Thanks, we have updated the description.
> 2. During our experiments we did vary our decoder, however the observation that direction information is increasing through the layers is because in later layers the direction information is easier to extract, or more “usable”. Indeed, by using a fixed decoder trained in the same conditions, we can evaluate how accessible this information is, an important difference from Shannon’s mutual information, which indeed could not be increasing through the layers. Practical considerations in choosing a decoder involved mitigating against overfitting (the case when using a very complicated decoder), but ensuring our decoder to be complex enough to capture the information. We achieved this by using relatively small and regularized networks.

---

### Official Review · AnonReviewer2 · 2020-10-28
**Interesting approach to understanding how representations form**

**Rating:** 7
**Confidence:** 3

**Review:**

Broadly, this work is an attempt to understand how neural networks can form generalizable representations while being severely overparameterized. This work proposes an information theoretic measure, called the "usable information", and use it to quantify the amount of relevant information in different layers of a neural network during training. The key idea is that, in order for the information represented in one layer to be "usable" by the next layer, it should be decodable by a simple transformation (affine + element-wise nonlinearity).

Pros:
- (significance) The "usable information" is a variant of mutual information, which replaces the expensive conditional entropy term with a cross-entropy loss that is more readily computable from a neural network. A computationally efficient measure for information has a potential for being a generally useful tool in the broad community.
- (quality) I like the approach of this study, which resembles how natural science tries to understand the function of a complex system (such as the brain, i.e., the real neural network) empirically. The choice of the task was also appropriate: it provides a good intuition about the relevant vs. irrelevant information, as well as a bridge to neuroscience studies that may lead to insightful discussion.
- (clarity) The paper is clearly written and easy to follow in most places.

Cons:
- (originality) One thing I expected to find in the paper was some review of other information theoretic measures that were proposed/used in the context of neural networks; proposing a cheaper alternative for the mutual information itself can't be a new idea (although if it is, that would be worth noting too). It would be fair to include a discussion along this line, and perhaps point out the properties of "usable information" that makes it particularly appealing.
- (clarity) The presentation of Fig 4 is not clear to me. (i) Which plots belong to which axis, and what is the third curve "val" in black? (ii) "a positive correlation with the minimality of the representation and generalization performances": this sounds vague. Can you quantify?

Additional comments:
- "Does SGD always lead to equivalent representations, or does SGD trace a path through parameter space that leverages structure present in the initialization?": I don't understand this sentence in the introduction. Also related, it would be nice to add a sentence or two to unpack the idea of "implicit regularization through SGD".
- In Fig 2d, why do the four marker types appear somewhat separated (although not by a large margin), e.g., red x, green o, green x then red o?
- The observation about non-random initialization is very interesting. In this example, keeping the old information does not seem to compromise the performance of the network in the current task. Do you think this is generally true for neural networks, or could there be a regime where retaining information about a previously relevant (but no longer relevant) information has an actual cost?

Overall, I think this is an interesting paper that presents a promising approach toward the understanding of how informative representations are formed by training, one of the most fundamental questions in deep learning.

**UPDATE:** Most of my questions/comments are addressed in the revised version of the paper and the author responses. I maintain my support for acceptance.

---

> ### Author Response · Authors · 2020-11-21
> **Response**
>
> We thank the reviewer for their comments and suggestions.
>
> > One thing I expected to find in the paper was some review of other information theoretic measures that were proposed/used in the context of neural networks; proposing a cheaper alternative for the mutual information itself can't be a new idea (although if it is, that would be worth noting too). It would be fair to include a discussion along this line, and perhaps point out the properties of "usable information" that makes it particularly appealing.
>
> We have updated the text. As was helpfully pointed out, the idea of using variational approximations to mutual information is not new. Rather, our paper proposes that it is a meaningful quantity to consider to study the training and behavior of a DNN, and how the training algorithm, hyper-parameters and pre-training can affect it. We have included further properties in that in addition to being a lower bound on Shannon’s mutual information, that it 1) is computable from samples, 2) that it is comparable throughout learning, and 3) that it not constrained to follow the data processing inequality, in contrast to mutual information, and is consistent with the representation learning view that subsequent representations are seen as extracting more useful features from the input through transformations.
>
> > The presentation of Fig 4 is not clear to me. (i) Which plots belong to which axis, and what is the third curve "val" in black? (ii) "a positive correlation with the minimality of the representation and generalization performances": this sounds vague. Can you quantify?
>
> We updated this plot, changing the validation accuracy to a green dashed line, as well as coloring the right hand side of the y-axis green to clarify the plot. The validation accuracy corresponds with the right hand axis, and the other curves are associated with the left hand axis. We have also added clarifications in text, and we hope that the updated version is more clear.
>
> We agree that this was potentially confusing and have opted to remove this sentence, instead to focus on the relationship between pretraining on color and minimality, as well as pretraining on color and generalization in the CB task, which is evident directly from the plot.
>
> > "Does SGD always lead to equivalent representations, or does SGD trace a path through parameter space that leverages structure present in the initialization?": I don't understand this sentence in the introduction. Also related, it would be nice to add a sentence or two to unpack the idea of "implicit regularization through SGD".
>
> We have clarified the text, and we have added an explanation for what we mean by implicit regularization, referring to the regularization from a large learning rate and small batch size.
>
> > In Fig 2d, why do the four marker types appear somewhat separated (although not by a large margin), e.g., red x, green o, green x then red o?
>
> This is due to random noise added to the color of the checkerboard input.
>
> > The observation about non-random initialization is very interesting. In this example, keeping the old information does not seem to compromise the performance of the network in the current task. Do you think this is generally true for neural networks, or could there be a regime where retaining information about a previously relevant (but no longer relevant) information has an actual cost?
>
> In general, we believe that to address the question of whether there is an effect of storing additional information in addition to the task information, one needs to consider not just the resulting solution but also the trajectory that led to the solution. Indeed, one can imagine a perfectly sufficient solution that is not minimal. However, in learning such a solution, old information affects the learning trajectory and may potentially interfere with learning a perfectly sufficient solution. This effect of old information on learning dynamics is  task and feature dependent. Further, we believe that representations that store additional information of the training set may be a signature of overfitting, as it is a sign that the network has learned qualities of the training distribution that are not necessary for test inference (as in Fig. 8).

---

### Official Review · AnonReviewer1 · 2020-10-28
**Interesting ideas but insufficient experimental evidence**

**Rating:** 3
**Confidence:** 4

**Review:**

This work introduces a notion of "usable information" in neural network representations (essentially, decodability of information by a neural network), and suggests that learned representations are "minimal" (discard task-irrelevant information) when training begins from random initialization, but not necessarily when beginning from other initializations.

Pros:

-- the definition of usable information is reasonable and likely useful for future analyses (though see below)
-- the questions addressed by the paper are interesting / important and are in need of thorough empirical study

Cons:

-- the tasks used in this paper are very simple, with even/odd MNIST classification being the hardest task considered (and most analysis is conducted on an even less complex task, which is similar to a simple XOR).  It is very hard to know whether the paper's conclusions would generalize to tasks of interest to the machine learning community, or even to other simple tasks with different structure

-- It is not clear from the pretraining experiments whether the negative results are due to pretraining or just the scale of the weights.  If networks were initialized randomly with mean / std taken from the pretrained network, would they also not learn minimal representations?

-- It seems that the results of the paper must necessarily depend on several hyperparameters which were not explored.  For instance, if the learning rate in early layers is set sufficiently small, the network should learn these simple tasks without minimal representations.

-- The result about generalization correlating with minimality was not confirmed on MNIST.  It is not clear whether this is because the result does not hold on MNIST or simply because the authors did not test it.

-- Transfer learning is known to be helpful in some practical settings.  Is this framework able to account for situations when transfer might be helpful, as well as harmful?  More discussion of this is needed


Overall, given that this is an empirical paper (no new theory is provided), it is important for the experiments to be extensive and comprehensive.  The experiments in this paper, though they touch on interesting ideas, are not thorough enough to convince a reader of the authors' broader claims.

---

> ### Author Response · Authors · 2020-11-21
> **Response (Part 1)**
>
> We thank the reviewer for their comments and suggestions.
>
> > the tasks used in this paper are very simple, with even/odd MNIST classification being the hardest task considered (and most analysis is conducted on an even less complex task, which is similar to a simple XOR). It is very hard to know whether the paper's conclusions would generalize to tasks of interest to the machine learning community, or even to other simple tasks with different structure
>
> We initially considered the Checkerboard Task as a simplified and easy to interpret setting in which to study the evolution of minimal sufficient representations, but we agree that it is critical to understand how the insights gained from the CB Task apply to realistic tasks and architectures. To this end (prompted also by R3), we have performed several new experiments with standard convolutional architectures (ResNet, AllCNN) on CIFAR-10 and CIFAR-100.
>
> Following R3’s suggestion, we train the network to predict a coarse labelling of the classes; in the CIFAR-100, this corresponded to the 20 superclasses, while in the CIFAR-10 case, we divided the samples in two super-classes, one containing the even classes, the other containing the odd classes. We then measured how much usable information the layers of the network have about the original fine-grained classes. Since the information about the specific fine-grained class is not needed to solve the task (it is nuisance information), we expected a minimal representation to not contain any such information. Indeed, we found that when training from a random initialization, networks increased the information about a semantically meaningful, but ultimately irrelevant nuisance information, and then later ``forgot’’ this information, ultimately arriving at a nearly minimal representation. On the other hand, when the learning rate is small or the batch size is large, we observe that, as expected, the network does not forget this information, and the representation does not become minimal. We discuss the full results in Sect. 4.2 and Sect. 4.3.  Overall, the use of more challenging tasks corroborated the intuitions gained using the simple CB task, allowing us to expose the learning dynamics in modern networks.
>
> > It is not clear from the pretraining experiments whether the negative results are due to pretraining or just the scale of the weights. If networks were initialized randomly with mean / std taken from the pretrained network, would they also not learn minimal representations?
>
> We tested this and found that in the CB task the scale of the weights led networks to learn non-minimal representations. In the CB task, an increased scale led to color separability in the output layer, since color information was directly present in the input at these scales. We note that this does not imply that learning an irrelevant feature does not affect subsequent training, but that the distribution of the initialization also affects the training dynamics. We comment on this in the text.
>
> > It seems that the results of the paper must necessarily depend on several hyperparameters which were not explored. For instance, if the learning rate in early layers is set sufficiently small, the network should learn these simple tasks without minimal representations.
>
> We thank the reviewer for bringing this up. We varied the hyperparameters in the ResNet trained on CIFAR-10, and indeed found that a small learning rate did not lead to minimal representations. This is shown in Fig 4c and Fig 8. We also varied the batch size, and found that only at small batch sizes (at 128 or less) did the network form a minimal representation, again suggesting that the noise from the stochasticity of SGD is crucial for these minimal representations. These findings corroborate that minimality of the representation depends on the noise of the training process. We note that networks with very low learning rates and networks with large batch sizes, which also did not learn minimal representations, had worse validation accuracy.

---

> ### Author Response · Authors · 2020-11-21
> **Response (part 2)**
>
> > The result about generalization correlating with minimality was not confirmed on MNIST. It is not clear whether this is because the result does not hold on MNIST or simply because the authors did not test it.
>
> We did not thoroughly investigate the generalization in MNIST, since due to the simplicity of the task, we did not observe a noticeable difference in the validation accuracy. On CIFAR-10, we find that for small learning rates and large batch sizes, where the representations are not minimal, that the generalization performance is worse, see for instance Fig 8, where the validation accuracy is 4.1 % lower than when using a larger learning rate. However, there does appear to be a regime where the representations are not minimal, but the validation accuracies are comparable (for example with a batch size of 256). Thus, we believe that our work provides a framework for addressing this foundational question as to the link between minimality and generalization carefully on realistic tasks and architectures. We do emphasize that under standard parameter settings of SGD, which have been shown empirically to lead to good generalization, had minimal representations. In future work, we will seek towards understanding whether minimal representations are necessary for good generalization, or rather an epiphenomenon, being a result of the successful training dynamics that learned important features early in training.
>
> > Transfer learning is known to be helpful in some practical settings. Is this framework able to account for situations when transfer might be helpful, as well as harmful? More discussion of this is needed
>
> We agree that transfer learning has been shown to be useful in practical settings. Ultimately it depends on the features learned, and whether they are helpful for learning the underlying task structure, or whether they lead the network to use greedy information in the initialization to learn a suboptimal global solution. In the case of CIFAR-100, pretraining on fine-labels improves the performance on coarse-grained classification, with validation accuracy increasing by 2.1%. This may be explained by the fact that fine-grained training helps the representation inglobate more useful features (that is, the representation is more sufficient). On the other hand, pretraining can learn features that are not useful for the target task but can be used to overfit (making the representation less minimal). We have added a short discussion about this in the paper. We believe this is an interesting avenue for future research.

---

### Official Review · AnonReviewer4 · 2020-11-03
**Studies minimality of neural network representations using a simple neuroscience-motivated task**

**Rating:** 7
**Confidence:** 4

**Review:**

The authors contribute to the recent research on whether neural network training (in particular, SGD) favors minimal representations, in which irrelevant information is not represented by deeper layers. They do so by implementing a simple neuroscience-inspired task, in which the network is asked to make a decision by combining color and target information. Importantly, the network's output is conditionally independent of the color information, given the direction decision, so the color information is in some sense irrelevant at the later stages. Using this, the authors quantify the 'relevant' and 'irrelevant' information in different layers of the neural network during training. Interestingly, the authors show that minimal representation are uncovered only if the network is started with random initial weights. Information is quantified using a simple decoder network.

The article is clearly written and has a simple (in a good way) and interesting message. However, I also have some criticisms, especially regarding the conceptual underpinnings.

When any neural network is predicting a deterministic function f : X -> Y, *all input features* are irrelevant to the output distribution when conditioned on the output itself. In other words, the minimal representation in a deterministic task is simply the output itself. (The situation is different when the task involves predicting a non-degenerate probability distribution P(Y|X), in which case the minimal representation -- i.e., the sufficient statistics -- can have an arbitrary amount of information.) In the information bottleneck community, this was mentioned in https://arxiv.org/pdf/1703.00810.pdf (section 2.4) and explored in https://arxiv.org/abs/1808.07593.

In motivating the paper, the authors appear to confuse two types of "irrelevant features":
	(1) when an input feature is useless for prediction, i.e., changing it does not change the predictions, and
	(2) when information about an input feature is independent of the output distribution, when conditioned on the output.
For a deterministic prediction task, all features type 2, but not all features are type 1. The authors have the following text:
      "We believe this task ... captures key structure from deep learning tasks. For example, in image classification, consider classifying an image as a car, which take on various colors. A representation in the last layer is typically conditionally independent of irrelevant input variations (i.e., the representation does not change based on differences in color)."
If I understand the example, this is building off the intuition that "color of car" is irrelevant because it is a type 1 feature (not useful for prediction). In fact, it can be conditionally independent because it is type 2. Moreover, in the authors' task "color of checkerboard" is not type 1 (it is very relevant for the output -- changing it changes the output) but it can also be conditionally independent (since it is type 2).

Given the above arguments, the degree to which features are conditionally independent in middle layers does not necessarily reflect how useful they are for prediction.

I have two other, more minor comments:
1) The notion of "direction information" is somewhat confusing, as one can think about two kinds of direction information: (1) information about which targets (i.e., directions) correspond to which colors (which is provided as part of the input), and (2) information about the final reaching direction (i.e., the output). Given the points made above, if I understand correctly, information about which targets correspond to which colors is just as irrelevant as the color information, when conditioned on the output. I would suggest referring to the second kind of information (the one mainly discussed in the paper) as "output information".
2) The authors should probably cite (and may be interested in) https://arxiv.org/abs/2009.12789 (NeuroIPS 2020), which also proposes to estimate mutual information using a practical family of decoders.

---

> ### Author Response · Authors · 2020-11-21
> **Response**
>
> We thank the reviewer for their comments and suggestions.
>
> > In motivating the paper, the authors appear to confuse two types of "irrelevant features": (1) when an input feature is useless for prediction, i.e., changing it does not change the predictions, and (2) when information about an input feature is independent of the output distribution, when conditioned on the output. For a deterministic prediction task, all features type 2, but not all features are type 1. The authors have the following text: "We believe this task ... captures key structure from deep learning tasks. For example, in image classification, consider classifying an image as a car, which take on various colors. A representation in the last layer is typically conditionally independent of irrelevant input variations (i.e., the representation does not change based on differences in color)." If I understand the example, this is building off the intuition that "color of car" is irrelevant because it is a type 1 feature (not useful for prediction). In fact, it can be conditionally independent because it is type 2. Moreover, in the authors' task "color of checkerboard" is not type 1 (it is very relevant for the output -- changing it changes the output) but it can also be conditionally independent (since it is type 2).
>
> We agree with the reviewer that our explanation was confusing. We have clarified the notion of relevant and irrelevant information in the text. We had initially attempted to clarify these notions in the original submission (what is now Section B.6) but had offered a confusing description in the main text. Indeed, since the information needs to be used, it needs to be represented at some stage, in contrast to purely irrelevant information.
>
> > The notion of "direction information" is somewhat confusing, as one can think about two kinds of direction information: (1) information about which targets (i.e., directions) correspond to which colors (which is provided as part of the input), and (2) information about the final reaching direction (i.e., the output). Given the points made above, if I understand correctly, information about which targets correspond to which colors is just as irrelevant as the color information, when conditioned on the output. I would suggest referring to the second kind of information (the one mainly discussed in the paper) as "output information".
>
> You understood correctly and we are considering making this change, though in the revised manuscript posted we have not yet made it. We hesitate since the output changes from 'direction' to 'color' during different parts of the paper, and we are afraid that simplifying the description to output information may cause more confusion.
>
> > The authors should probably cite (and may be interested in) https://arxiv.org/abs/2009.12789 (NeuroIPS 2020), which also proposes to estimate mutual information using a practical family of decoders.
>
> Thank you for pointing out this reference; indeed it is relevant, and we have included this in the related work section.

---

### Public Comment · ~Andreas_Kirsch1 · 2020-11-10
**Prior literature for usable information in a presentation**

### Important prior literature

I would like to point towards a few references that are not mentioned in this paper but are relevant and important prior art for usable information under computational constraints:

[Xu, Yilun, et al. "A Theory of Usable Information under Computational Constraints." International Conference on Learning Representations. 2019](https://arxiv.org/abs/2002.10689) introduce $\mathcal{V}$-Information, which is defined as
$$ I_{\mathcal{V}}(X \rightarrow Y)=H_{\mathcal{V}}(Y)-H_{\mathcal{V}}(Y \mid X) $$
which is "entropy - cross-entropy" restricted to a model family $\mathcal{V}$, so the usable information also introduced in this paper. It is somewhat more general though as it does not require the LHS term to be exact like in this paper, but it is also estimated within the model family.

Moreover, [Dubois, Yann, et al. "Learning Optimal Representations with the Decodable Information Bottleneck." Advances in Neural Information Processing Systems 33 (2020)](https://arxiv.org/pdf/2009.12789.pdf) use $\mathcal{V}$-Information for examining an Information Bottleneck objective using this information instead of regular entropy/mutual information.

Finally, [McAllester, David, and Karl Stratos. "Formal limitations on the measurement of mutual information." International Conference on Artificial Intelligence and Statistics. 2020.](https://arxiv.org/abs/1811.04251) also examines
$$ I\left(X, Y ; p_{X Y}\right)=\inf_{q_{X}} H\left(p_{X}, q_{X}\right) -\inf_{q_{X \mid Y}} H\left(p_{X \mid Y}, q_{X \mid Y}\right) $$
as a mutual information estimate, which is not upper-bounded by the sample count logarithm like MINE/DV variants.

### Average test loss/average log-likelihood as usable information

In this paper, $H(Y)$ is known and does not have to be estimated (which is a more specific case than the ones above). The usable information estimate is thus a shifted average log-likelihood for auxiliary networks that try to decode different information from intermediate layers. In other words, the experiments analyze the test loss when training auxiliary heads with cross-entropy to decode the information from (intermediate) layers.

---

> ### Author Response · Authors · 2020-11-21
> **Response**
>
> Thank you for pointing us towards these references. After we submitted our ICLR submission, we became aware of Dubois et al. (2020) [which was posted on arXiv a day before our ICLR submission and was also helpfully pointed out by R4], as well as the reference therein to Xu et al (2020). These works, especially that of Xu et al (2020) focus on developing an information-theoretic theory of variational approximations to information. The idea of using a linear classifier or a simple network to measure information is pretty standard. Rather, our focus is to use a notion of usable information and show how it differs layer by layer, and critically how it evolves during the training of the network, both when training in standard conditions and when perturbing the initial part of the training process. We also show how it changes based on the hyperparameters. All of this is orthogonal to the theory in the paper cited. We now discuss the work in the related work section.

---

### Author Response · Authors · 2020-11-21
**Revision uploaded**

We thank all the reviewers for their thoughtful comments and experimental suggestions. We believe their feedback has led to a significantly improved manuscript, now posted. In particular, we have added several new experiments on standard tasks (CIFAR-10 and CIFAR-100, as suggested by R3 and R1) using modern architectures. We observe interesting learning dynamics, where the network increases information about semantically meaningful but ultimately irrelevant information early in training, discarding it later on to learn a minimal sufficient representation (Section 4.2). We also evaluate how these dynamics depend on hyperparameters, network architecture and task (Section 4.3). We observe these learning dynamics only when batch sizes are small and learning rates are large, consistent with the notion that the implicit regularization from SGD is crucial for the formation of optimal representations. We have restructured the experiments section to highlight these findings in the main text.

---

### Decision · Program_Chairs · 2021-01-07
**Final Decision**

**Decision:**

Accept (Poster)

**Comment:**

This paper proposes that we can understand the evolution of representations in deep neural networks during training using the concept of "usable information". This is effectively an indirect measure of how much information the network maintains about a given categorical variable, Y, and the authors show that it is in fact a variational lower bound on the amount of mutual information that the network's representations have with Y. The authors show that in deep neural networks the usable information that is maintained for different variables during training depends on the task, such that task irrelevant variables (but not task relevant variables) eventually have their usable information reduced, leading to "minimal sufficient representations".

The initial reviews were mixed. A common theme in the critiques was the lack of evidence of the generalization and scalability of these results. The authors addressed these concerns by including new experiments on different architectures and the CIFAR datasets, leading one reviewer to increase their score. The final scores stood at 3, 7 ,7, 7. Given the overall positive reviews, interesting subject matter, and relevance to understanding learned representations in deep networks, this paper seems appropriate for acceptance in the AC's opinion.